# Warming and trophic structure tightly control phytoplankton bloom amplitude, composition and succession

Thomas Trombetta[1]*, Behzad Mostajir[1], Justine Courboulès[1], Maria Protopapa[2], Sébastien Mas[3], Nicole Aberle[4,5], Francesca Vidussi[1]*

1 MARBEC (Marine Biodiversity, Exploitation and Conservation), Univ Montpellier, CNRS, Ifremer, IRD, Montpellier, France, 2 HCMR (Hellenic Centre for Marine Research), Institute of Oceanography, Anavissos, Greece, 3 MEDIMEER (Mediterranean Platform for Marine Ecosystems Experimental Research), OSU OREME, CNRS, University Montpellier, IRD, INRAE, Sète, France, 4 Department of Biology, NTNU (Norwegian University of Science and Technology), Trondhjem Biological Station, Trondheim, Norway, 5 Institute of Marine Ecosystem and Fisheries Science (IMF), Universität Hamburg, Hamburg, Germany

* thomas.trombetta@gmail.com (TT); francesca.vidussi@cnrs.fr (FV)

**Data Availability Statement:** All relevant data are within the manuscript and its Supporting Information files.

## Abstract

To better identify the responses of phytoplankton blooms to warming conditions as expected in a climate change context, an *in situ* mesocosm experiment was carried out in a coastal Mediterranean lagoon (Thau Lagoon, South of France) in April 2018. Our objective was to assess both the direct and indirect effects of warming on phytoplankton, particularly those mediated by top-down control. Four treatments were applied: 1) natural planktonic community with ambient water temperature (C); 2) natural planktonic community at +3°C elevated temperature (T); 3) exclusion of larger zooplankton (> 200 μm; mesozooplankton) leaving microzooplankton predominant with ambient water temperature (MicroZ); and 4) exclusion of larger zooplankton (> 200 μm; mesozooplankton) at +3°C elevated temperature (TMicroZ). Warming strongly depressed the amplitude of the phytoplankton bloom as the chlorophyll *a* concentration was twice lower in the T treatment. This decline under warmer conditions was most likely imputed to increase top-down control by zooplankton. However, removal of mesozooplankton resulted in an opposite trend, with a higher bloom amplitude observed under warmer conditions (MicroZ vs. TMicroZ) pointing at a strong interplay between micro- and mesozooplankton and the effect of warming for the spring phytoplankton blooms. Furthermore, both warming and mesozooplankton exclusion induced shifts in phytoplankton community composition during bloom and post-bloom periods, favoring dinoflagellates and small green algae at the expense of diatoms and prymnesiophytes. Moreover, warming altered phytoplankton succession by promoting an early bloom of small green flagellates, and a late bloom of diatoms. Our findings clearly highlighted the sensitivity of phytoplankton blooms amplitudes, community composition and succession patterns to temperature increases, as well as the key role of initial zooplankton community composition to elicit opposite response in bloom dynamics. It also points out that warmer conditions might favor dinoflagellates and small green algae, irrespective of zooplankton community composition, with potential implications for food web dynamics and energy transfer efficiency under future ocean condition.

**Funding:** This study was part of the Photo-Phyto project funded by the French National Research Agency (ANR-14-CE02-0018) financing also the PhD of TT. The experiment was opened to transnational access throughout the AQUACOSM project (European Union's Horizon 2020 research and innovation program H2020/2017-2020 under grant agreement n°731065.) which financed NAM, and MP participation. Microscopy and cytometry equipment were provided by the MICROBEX platform of MARBEC/CeMEB LabEX with the support of LabEx CeMEB, an ANR "Investissements d'avenir" program (ANR-10- LABX-04-01). The funders had no role in study design, data collection and analysis, decision to publish, or preparation of the manuscript.

**Competing interests:** The authors have declared that no competing interests exist.

## Introduction

Phytoplankton blooms in temperate and subpolar regions are major phenomena supporting the productivity of these ecosystems [1]. Blooms in coastal areas are especially crucial because they contribute substantially to the annual primary production and energy transfer to the upper trophic levels. Such blooms support the entire marine food web in areas that cover only 6% of the world's surface but provide between 22% and 43% of its ecosystem services [2,3].

Global warming models predict that sea-surface temperature will continue to increase by 1.0–5.7°C by 2080 [4], which is likely to affect phytoplankton bloom production at the base of the food web. However, warming can have opposing effects on phytoplankton blooms depending on specific characteristics of an ecosystem and a given study period [5]. Generally, bloom amplitude or frequency increases in relation to warming are attributed to enhanced upwelling, phytoplankton metabolism, and extension of thermal niches [5–7], while its reduction is attributed to a weakening of upwelling events, stratification, faster nutrient depletion, or enhanced consumer control [5,8–10].

A recent *in situ* study highlighted that warming occurring in shallow coastal waters, especially during the winter, can also reduce spring phytoplankton biomass accumulation, shifting the community toward the dominance of smaller phytoplankton species without observable links to nutrient inputs [11]. This suggested that warming might have a more complex effect, probably not solely attributable to the modification of abiotic factors in the water column, but also to the role of overwintering consumers [11]. Experimental studies revealed that interspecific differences in the degree of thermal tolerance and the capacity for adaptation might be the key to understand changes in bloom amplitude and community composition under future warming scenarios [12–14]. Models based on the metabolic theory of ecology and experimental studies have highlighted that warmer conditions are more favorable to heterotrophic than autotrophic processes [15–17], indicating an increasing role of grazers as bloom modifiers and controllers under warming. Furthermore, indoor mesocosm experiments have demonstrated a strengthening of both the meso- and microzooplankton top-down control on phytoplankton with increasing temperature [13] suggesting that phytoplankton responses vary depending on zooplankton size and predation pressure. However, there is a need to study the response of natural phytoplankton communities to warming in more detail, particularly direct (physiological) effects on phytoplankton as well as indirect effects (e.g., in relation to grazing). Since warming affects different trophic levels unequally, it can alter the trophic cascade and lead to changes in e.g., bloom magnitude, timing, succession and composition. Especially microbial processes, such as protozoans' growth and grazing, are increased by warming, affecting both their main preys and predators (phytoplankton and metazooplankton) biomass and composition, due to changes in energy transfer efficiency [18,19]. The idea to exclude mesozooplankton (> 200 μm), in combination with warming, can allow to better disentangle the role of microzooplankton (< 200 μm) including that of protozooplankton and its interplay with temperature on phytoplankton blooms.

In this study, we conducted a factorial *in situ* mesocosm experiment during a spring bloom with a natural plankton community from a shallow, productive coastal lagoon in the North-Western Mediterranean Sea (Thau Lagoon in southern France), where the factors water temperature and zooplankton size spectra (i.e., trophic cascade structure) were manipulated. Two previous studies on this experiment, focusing only on the elevated temperature treatment, reported decreases in the phytoplankton biomass during the bloom under warming and, in oxygen production [20,21]. However, these studies focused solely on the dynamics within the microbial loop (bacteria and small phytoplankton) or on the functioning of the planktonic community while the role of predator dynamics and trophic cascades were not considered in

detail. The objectives of the present study were to investigate phytoplankton bloom dynamics at ambient and elevated temperature conditions, as well as direct and indirect effects on phytoplankton standing stocks and community composition mediated by microzooplankton (< 200 μm) or by both micro- and mesozooplankton (> 200 μm). We hypothesized that both warming and removal of mesozooplankton reduces phytoplankton bloom amplitudes due to an increase of top-down pressure. Further, we hypothesized that the response of phytoplankton to warming is tightly linked to the initial zooplankton community structure (zooplankton cut-offs < 200 μm and > 200 μm) as the interaction between temperature and zooplankton structure induces trophic cascade that lead to contrasting response in bloom amplitudes, timing and succession.

## Material and methods

### Experimental design

The *in situ* mesocosm experiment was carried out in the Thau Lagoon, at the MEDIterranean platform for Marine Ecosystem Experimental Research (MEDIMEER). Thau Lagoon is a shallow coastal lagoon (mean depth: 4 m), located in the northwestern Mediterranean shore (43˚ 24'00" N, 3˚36'00" E) and experiences a wide range of temperatures from 4–29˚C. The Thau Lagoon is a productive Mediterranean lagoon that hosts oyster farms. It is a mesotrophic lagoon, but phytoplankton can sometimes experience phosphorus and nitrogen limitations [22]. The mesocosm bags were composed of 200 μm thick mixed vinyl acetate polyethylene film (Insinööritoimisto Haikonen Oy), which transmitted 53% of the ultraviolet B radiation (UVBR) and 77% of the photosynthetically available radiation (PAR: 400–700 nm) [23]. To prevent any contamination from rain or other external inputs, the mesocosms were covered with removable and opening domes, composed of the same transparent polyethylene, still allowing gas exchange with the atmosphere. The mesocosms were 3 m high and 1.2 m wide and were held 1 m above the water surface by floating structures; thus, the mesocosm water column depth was 2 m and contained 2200 L of Thau Lagoon water. The mesocosm water column was gently and permanently homogenized with an immersed pump (Model 24, 12V, Rule) to simulate mixing. A daily turnover time of ca. 3.5 $d^{-1}$ was set that was adjusted for each mesocosm at the beginning of the experiment and maintained throughout the experiment. No additional oxygenation of the mesocosms was applied throughout the experiment. Mesocosms were moored in situ at the pontoon of MEDIMEER, where the depth was approximately 3 m. The in situ mesocosm experiment lasted 19 days from April 5 to 23, 2018.

### Exclusion of mesozooplankton and increase in seawater temperatures

Fourteen mesocosms were filled on April 5, 2018 (hereafter called day 0). Twelve mesocosms were used for treatments (see below) and two for incubation experiments (not presented in the study). The lagoon water was pumped (SXM2/A SG, Flygt) at a 1.5 m depth near the MEDIMEER pontoon and then screened through a 1000 μm mesh to remove large particles and debris into a large water pool container. For this study, four treatments of two factors (temperature × zooplankton community) were applied in triplicates: 1) ambient seawater temperature × natural planktonic community < 1000 μm (Control: C), 2) elevated seawater temperature (Δ+ 3˚C) × natural planktonic community < 1000 μm (T); 3) ambient seawater temperature × natural plankton community < 200 μm (MicroZ; mesozooplankon removal); and 4) elevated seawater temperature (Δ+ 3˚C) × natural plankton community < 200 μm (TMicroZ; mesozooplankon removal). Mesocosms were filled simultaneously through individual filling tubes with 1000 μm screened lagoon water (C and T) or with the same 1000 μm screened water passing through a 200 μm mesh size tissue (NITEX, SEFAR) tightly attached at

the end of the filling tubes to remove mesozooplankton (> 200 μm) in order to retain only microzooplankton fractions (< 200 μm) (MicroZ and TMicroZ).

Seawater temperature for the heated treatments (T and TMicroZ) was increased using a submersible heating element (Galvatec) immersed vertically at a depth of 1 m [23,24]. The ΔT between the heated and control treatments was controlled continuously by an automated system based on a closed-loop regulation according to [23]. To maintain the ΔT between the control and heated mesocosms, the seawater temperature was monitored at depths of 0.4, 0.8, and 1.2 m every 5 min. using thermistor probes (Campbell Scientific 107) in all mesocosms. The mean temperature of the heated mesocosms at the three depths was then compared to the mean temperature of the control mesocosms using a Campbell Scientific data logger (CR1000). The heating elements started or stopped automatically to reach or maintain the target ΔT value (for more detail, see [23]). To avoid a thermal stress to the organisms, the ΔT in both heated treatments (T and TMicroZ) was increased in two steps: an increase of 1.5˚C attained between days 1 and 2 followed by 1.5˚C between days 2 and 3 to reach the desired +3˚C relative to the natural water temperature of control treatments (this was maintained until the end of the experiment). We chose 1.5˚C per day as this is within the natural daily range of temperature variation in the Thau Lagoon [11] and therefore would not thermally shock the plankton community.

## Physical, chemical, and biological sampling

Mesocosm seawater temperature was monitored at mid-depth every day between 9:00 and 10:00 a.m. with a multiparameter sensor (EC300, VWR International) from day 2 until the end of the experiment. Due to poor weather conditions, the mesocosm seawater temperatures were not monitored on days 6 and 7. Thus, seawater temperature data from the thermistor probes at 9:00 a.m. were used that day to replace the missing data. Incident photosynthetically active radiation (PAR, 400–700 nm) was recorded at a high frequency (every 15 min.) using a Professional Weather Station (METPAK PRO, Gill Instruments) with a PAR sensor (Skye Instruments) installed on the pontoon. PAR monitoring started from day 6 (no data available before day 6 due to technical issues) at 6:45 p.m. until the end of the experiment. To identify the daily incident PAR dose received at the pontoon, the daily light integral (DLI) or daily dose was calculated using **Eq 1**:

$$\mathrm{DLI} = \sum PAR \times \Delta t \tag{1}$$

where $\Sigma$ *PAR* is the sum of the PAR measured during the day (96 measurements), and $\Delta t$ is the time interval between two measurements (900 s). As data from days 6 and 18 were incomplete, the DLI for these days was not calculated. Light penetration (PAR, 400–700 nm) was also measured in the mesocosms between 2:00 and 5:00 p.m. with a spectroradiometer (TriOS RAMSES ACC hyperspectral) and data are shown in **S1 Table in S1 File**. The near-surface diffuse attenuation coefficient (Kd) [25] was calculated using **Eq 2**:

$$\mathrm{Kd} = \frac{1}{Z} \times \ln\left(\frac{E_0}{E_Z}\right) \tag{2}$$

where z is the depth, $E_0$ the irradiance at below the surface and $E_Z$ the irradiance at Z depth.

Samples (5 L) for nutrient analyses were taken daily using a Niskin bottle between 8:30 and 9:30 a.m. for nutrient analyses, except for day 6 due to inclement weather.

To estimate the biological parameters, mesocosm seawater samples were taken daily between 8:30 and 10:00 a.m. at mid-depth (1 m) using a low vacuum pump in vacuum-resistant plastic bottles, which we then stored in a polycarbonate jerrycan pre-washed with acid

and rinsed with distilled water. Sub-sample aliquots were immediately taken from the samples for further analysis.

Microzooplankton ($< 200$ μm; microprotozooplankton and micrometazooplankton) was sampled by taking 100–200 mL of seawater from each treatment from the corresponding polycarbonate jerrycans. The samples were transferred to brown glass bottles and fixed with acidic Lugol's iodine solution (final concentration: 2%). Microzooplankton was sampled at 15 time points (days: 0, 1, 2, 3, 5, 7, 8, 9, 11, 13, 14, 15, 16, 17, 18) of which 14 sampling points were analyzed (days: 0, 2, 3, 5, 7, 8, 9, 11, 13, 14, 15, 16, 17, 18).

To determine the dominant mesozooplankton taxa, sampling of all mesocosms were made at day 0, 7, 14 and 18. A low-vacuum pump was used to sample 40 L from the polycarbonate jerrycan pre-washed with acid and rinsed with distilled water. Water samples were screened on a 20 μm plankton net (Hydrobios) to check eggs and juveniles ($< 200$ μm) and their growth during the experiment. The resulting samples were fixed with formaldehyde (4% final concentration) and stored until analysis.

## Nutrients analyses

To determine nitrite ($NO_2^-$), nitrate ($NO_3^-$), phosphate ($PO_4^{3-}$), and silicate ($SiO_2$) concentrations, two 13 mL aliquots were filtered through PP 0.45 μm filters (25 mm, Agilent Technologies) that were pre-washed three times and stored at –20°C until analysis. Samples were analyzed using a continuous flow analyzer (San$^{++}$, Skalar) following standard nutrient analysis methods [26]. To measure ammonium ($NH_4^+$) concentrations, 50 mL subsamples collected in Niskin bottles were taken and were determined using the fluorometric method [27].

## Phytoplankton pigment analyses and bloom identification

Sub-samples of 1 L were taken from the 10 L jerrycans to determine phytoplankton pigment concentrations. Sub-samples were filtered through glass-fiber filters (GF/F Whatman: 25 mm, nominal pore size: 0.7 μm) and stored at -80°C until analysis. Pigment concentrations were measured using high-performance liquid chromatography (HPLC, Waters) following the method described by [28], with some modifications to adapt it to the HPLC system used (i.e., analysis time of 45 min.) and the extraction protocol described by [24].

To distinguish the different phytoplankton phases of the experiment, chlorophyll *a* (Chl *a*) concentration was used as a proxy for phytoplankton biomass. Bloom periods were identified by estimating the net phytoplankton growth rate using the phytoplankton biomass gain and loss [29]. Then, the phytoplankton daily Chl *a* net growth rate (r) was calculated for each mesocosm using Chl *a* concentration (**Eq 3**) and a daily mean and range of observations (minimum and maximum value) of the growth rate was calculated for each treatment.

$$r = \frac{\ln\left(\frac{C_{n+1}}{C_n}\right)}{t} \tag{3}$$

where $C_n$ is the daily Chl *a* concentration. A bloom was identified as a period of at least two consecutive days with a positive growth rate that ended after one day of negative growth rate [11]. This method allows the comparison of bloom periods without taking into account Ch *a* concentration threshold, as it can be highly variable from one environment to another. Increase in incident light from the mid-term experiment might have led to underestimation of the phytoplankton biomass and net growth rates due to photoacclimation [30]. However, as it occurred in all mesocosms and the main goal was to compare the phytoplankton response to different treatments, such potential underestimation could not be crucial.

**Table 1. Taxonomic pigments, their abbreviations, and corresponding taxa [31,33,34].**

| Pigments | Abbreviation | Presence in taxa |
|---|---|---|
| Chlorophyll *b* | Chl *b* | Green flagellates |
| Prasinoxanthin | Prasi | Prasinophytes |
| Zeaxanthin | Zea | Cyanobacteria, Chrysophytes |
| Alloxanthin | Allo | Cryptophytes |
| 19'hexanoyloxyfucoxanthin | 19HF | Haptophytes (Prymnesiophyceae) |
| Fucoxanthin | Fuco | Diatoms |
| Peridinin | Peri | Dinoflagellates |

In addition to Chl *a*, some pigments are representative of specific taxa [31,32]. Therefore, if the pigments investigated are wisely chosen, the whole phytoplankton diversity can be captured. For statistical analysis, we chose seven taxonomic pigments according to their significance as proxies of functional taxonomic groups, thus representing the taxonomic diversity of phytoplankton, and indicated their relevance as diagnostic pigments according to their description in the literature [31,33]. The seven major taxonomic pigments observed during the experiment and their taxonomic significance reported in the literature are presented in **Table 1**.

## Phytoplankton cytometric abundances

To quantify daily phytoplankton abundance (cyanobacteria, picophytoeukaryotes and nanophytoeukaryotes) by flow cytometry (FCM), 1.5 mL subsamples were taken from the 10 L jerrycans. Samples were fixed using 60 μL of glutaraldehyde (Grade1) and frozen in liquid nitrogen. Samples were stored at –80˚C until analysis. Phytoplankton analyses were performed using a CytoFLEX (Beckman Coulter) for 3 min. at high speed (60 μL min.$^{-1}$). Cytometry fluorescent beads (Polysciences, Inc.) of 1, 2, 6, 10 and 20 μm in diameter were added into the samples to estimate cell sizes. In addition, Trucount$^{TM}$ beads were added to accurately estimate the sample volume (BD Biosciences). Cyanobacteria were identified and enumerated based on their forward scatter (FSC) and natural phycoerythrin fluorescence (FL2; between 542 nm and 585 nm). Picophytoeukaryotes and nanophytoeukaryotes were identified and enumerated based on their FSC and natural chlorophyll fluorescence (FL3; 650 nm). One group of cyanobacteria (Cyano) was identified, one group of picophytoeukaryotes (Picoφ) between 1 μm and 2 μm, and several groups of nanophytoeukaryotes, which were pooled into one group (Nanoφ) between 2 μm and 20 μm.

## Microprotozooplankton and micrometazooplankton abundances

Microzooplankton (< 200 μm; microprotozooplankton and micrometazooplankton) abundance and community composition were analyzed using an inverted microscope (Utermöhl 1958) at 200x magnification and species identification was made to the lowest taxonomic level possible. Microprotozooplankton included ciliates (aloricate and loricate) and heterotrophic dinoflagellates (thecate and athecate), micrometazooplankton included veliger larvae, rotifers, nauplii, gastropod larvae, polychaeta larvae and trochophora larvae that were < 200 μm. Depending on microzooplankton abundance, either 50- or 100-mL settling chambers were used and samples were settled for at least 24 hours. Depending on abundance, either the half or full area of the bottom plate was counted for reliable abundance estimates.

## Dominant phytoplankton and mesozooplankton taxa

To determine the dominant phytoplankton taxa (> 5 μm), 110 mL subsamples were taken from the 10 L jerrycans and fixed with 4 mL of formaldehyde in a 125 mL amber glass stored

at 4˚C until microscopic analyses. 25 mL and 50 mL subsamples were taken from the 125 mL amber glass and settled for 24h in Utermöhl chambers and observed under an inverted microscope (Olympus IX-70). Dominant phytoplankton taxa were identified in the samples on days 0, 3, 9, 13, and 18. Identification was made using phytoplankton taxonomic keys [35,36].

Samples to identify the dominant mesozooplankton taxa abundance were analyzed under a stereomicroscope (Motic SMZ 717) in the samples on days 0, 7, 14, and 18. Identification was made using zooplankton taxonomic key [37].

## Statistical analysis

After the end of the experiment, once all mesocosm bags had been removed and carefully inspected in the workshop, some holes were observed in one of the TMicroZ mesocosms (most probably due to a storm that occurred between days 6 and 7) thus leading to suspect leaks. Therefore, this mesocosm was omitted from the data analyses. In addition, data analyses of three replicate mesocosms from the three other treatments showed significant deviations from the other replicates. Therefore, the functioning of the pumps used to ensure the homogenization of the mesocosm water was checked after the experiment and a dysfunction detected. Thus, the data of deviating replicate mesocosms was omitted retrospect and only two replicates of each treatment considered for data analysis hereafter. Therefore, all results presented here are the mean of duplicate mesocosms per treatment with their range of the observations (minimum and maximum values).

To test the effect of the different treatments (C, MicroZ, T, and TMicroZ) two-way repeated measures analyses of variance (RM-ANOVA) [38] were conducted on the concentrations of nutrients, Chl *a*, and taxonomic pigments, as well as on pico- and nanophytoplankton abundances. Two factors of two levels (temperature: ambient temperature and elevated temperature $\Delta+ 3˚C$; zooplankton community: natural planktonic community $< 1000$ μm and natural planktonic community $< 200$ μm) with interaction were tested and 'days' were set as random effect to prevent spurious autocorrelation effects. Normality was tested using the Shapiro-Wilk test and homogeneity of variance was assessed using the Levene test [39,40]. When assumptions were not meet despite logs, squared roots or Box-Cox transformations, Kruskall-Wallis tests were performed instead [41]. When significant, the comparison tests were followed by post hoc pairwise tests (Tukey's tests for RM-ANOVA and Wilcoxon tests for Kruskall-Wallis) to identify the effects of the different treatments [42,43]. Comparison tests were conducted separately on the three different phytoplankton bloom periods observed during the experiment ('pre-bloom', 'bloom', and 'post-bloom'; see the bloom period identification in the Result section) as effects were susceptible to inducing different responses on the various studied variables during these periods. As the bloom periods could have occurred at different times depending on the treatment, unified 'pre-bloom', 'bloom' and 'post-bloom' periods were identified using only the days of bloom period that were common among all treatments. As stated above, in this paper, we present two replicates of each treatment are presented and the "±" symbols in the text and related figures signify the range of the observations. Results of RM-ANOVA are presented in **S1 to S4 Figs in S1 File**. To highlight shifts in phytoplankton community composition between periods and treatments, redundancy analysis (RDA) was performed [44]. ANOVA for RDA using the vegan package for R was then used to identify the environmental drivers of these community shifts. PERMANOVA were also conducted between treatments and periods to assess phytoplankton community composition differences, using Bray-Curtis distance on pigments and cytometry composition [45]. All statistical analysis were performed using R software version 4.4.0 (The R Foundation).

## Results

### Temperature and DLI dynamics

At the beginning of the experiment, seawater temperature in the unheated treatments (C and in MicroZ) was 13.2°C on day 2 (**Fig 1**). The Δt of +3°C in the heated treatments (T and TMicroZ) was attained between days 2 and 4. The ambient seawater temperature was almost stable in all treatments from the beginning until day 10. Temperature variations were well mimicked by the heated treatments. Then, the water temperature increased naturally from days 10 to 17 to reach a maximum means of 18.4°C and 21.2°C in the control and heated treatments, respectively. Daily Light integral (PAR DLI) (**Fig 1**) at the surface water was initially low, particularly corresponding to the storm that occurred between days 7 and 9 (means of 19 ± 6 mol quanta m$^{-2}$ d$^{-1}$). The PAR DLI increased on day 10, remaining high until the end of the experiment, with a mean of 52 ± 3 mol quanta m$^{-2}$ d$^{-1}$.

### Chl *a* dynamics and bloom identification

Bloom periods were identified by estimating the net phytoplankton growth rate (**Fig 2**) as detailed in the Materials and Methods. At the beginning of the experiment, the Chl *a* concentrations were relatively low in all treatments (mean 0.84 ± 0.04 μg L$^{-1}$; **Fig 2**). In the control, a period of several consecutive positive daily Chl *a* net growth rates, indicating a bloom, occurred from days 2 to 10 (**Fig 2A**). It reached a maximum Chl *a* concentration on day 10 (4.51 ± 0.14 μg L$^{-1}$). The bloom period was followed by a post-bloom period, characterized by several consecutive negative daily growth rates (or close to zero) until the end of the experiment. A similar trend was observed when mesozooplankton was removed (MicroZ) but with a lower net growth rate during the bloom, leading to Chl *a* concentration peak at day 9 reduced

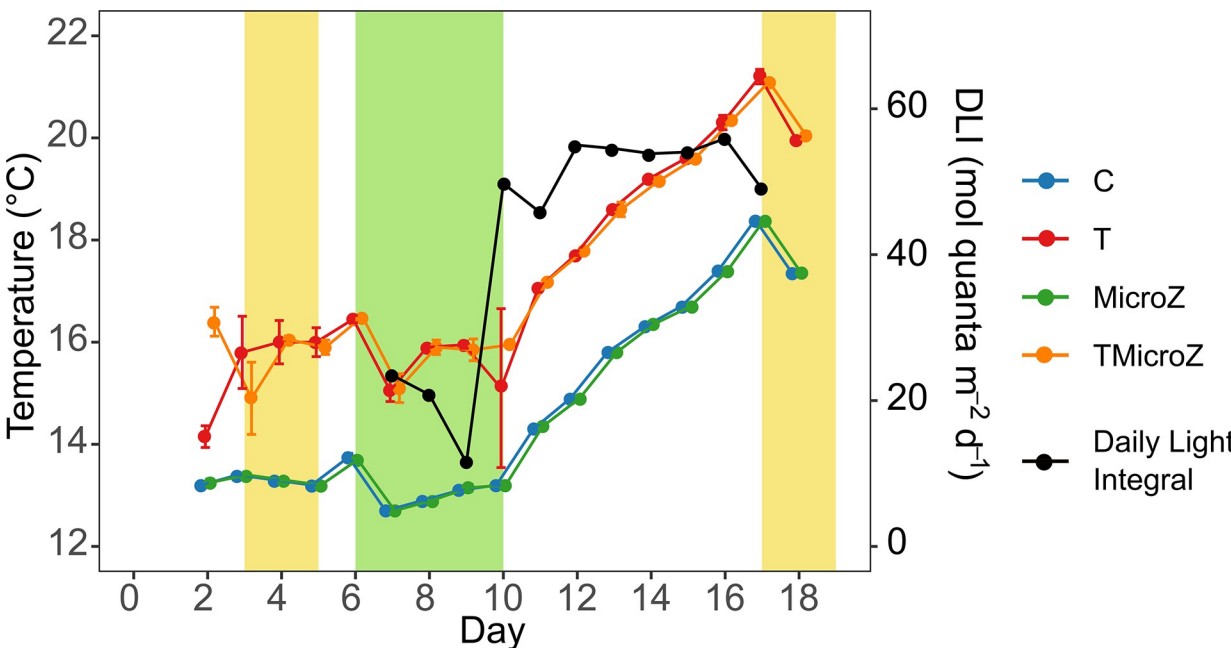

**Fig 1. Daily seawater temperature means (± range of the observations) for the different treatments and Daily Light Integral (DLI) dynamics.** C, control; T, treatment with water heated +3°C; MicroZ, mesozooplankton exclusion treatment; TMicroZ, water heated t +3°C and mesozooplankton exclusion treatment. Green background indicates the main bloom period common to all treatments and yellow background indicates the early and late blooms identified in T and TMicroZ treatments.

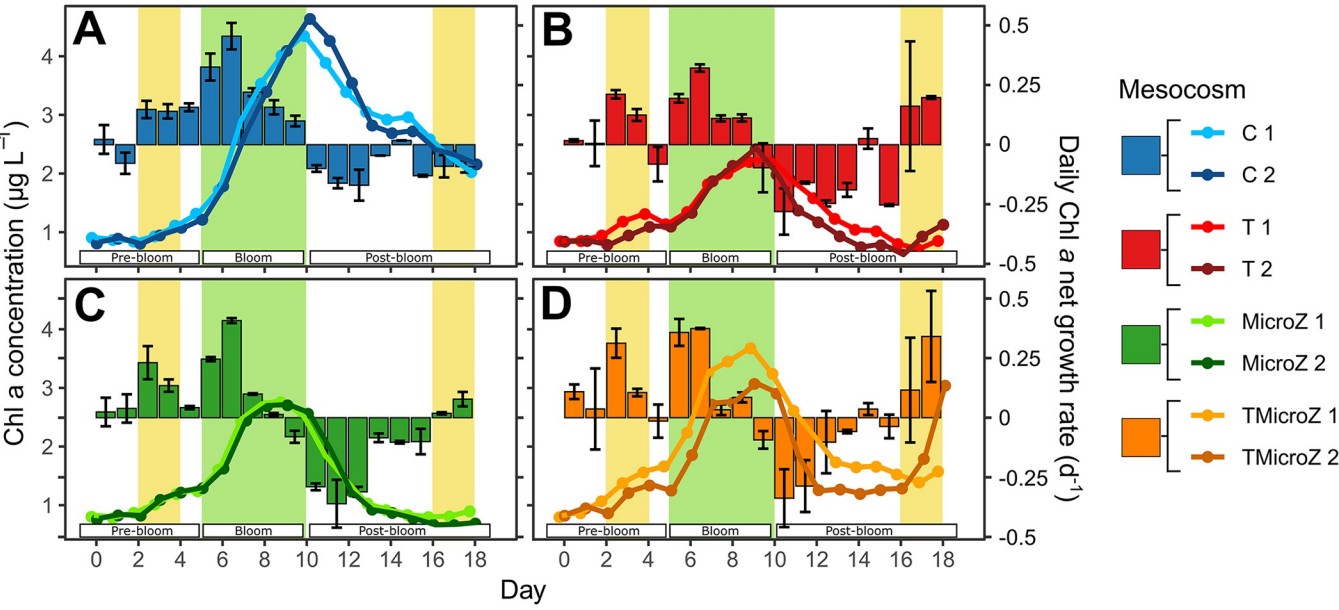

**Fig 2. Chl *a* concentrations and daily Chl *a* net growth rates (r) mean (±range of the observations) in the different treatments.** Lines and dots represent Chl *a* concentrations. Bars represent the daily Chl *a* net growth rates: (A) control (C) treatment, (B) heated +3˚C (T) treatment, (C) the mesozooplankton exclusion treatment (MicroZ treatment), and (D) the heated +3˚C and mesozooplankton exclusion treatment (TMicroZ). Green background indicates the main bloom period common to all treatments and yellow background indicates the early and late blooms identified in T and TMicroZ treatments. Pre-bloom, bloom, and post-bloom periods, separated by dotted lines, are the periods common in all treatments used for statistical analysis.

by 39% (2.75 ± 0.03 μg L$^{-1}$). Heated treatments (T and TMicroZ) showed high growth rates on days 2 and 3, followed by a day of negative growth (day 4) (**Fig 2B and 2D**). This period was identified as an early bloom, which ended on day 4. The main bloom period occurred from days 5 to 10 (peaking at day 9), with lower growth rates in the heated treatments than those in C. Warming reduced the Chl *a* concentration peak during the bloom by half (T: 2.33 ± 0.11 μg L$^{-1}$ vs. C: 4.51 ± 0.14 μg L$^{-1}$), while in the treatments without mesozooplankton, elevated temperature conditions increased the peak by 23% (TMicroZ: 3.39±0.31 μg L$^{-1}$ vs. MicroZ: 2.75 ±0.03 μg L$^{-1}$). The bloom was followed by a period of negative growth rate until days 14–16 (post-bloom period). During the last two days, positive growth rates were observed in the non-control treatments (T, MicroZ and TMicroZ), which could be considered the late-bloom period. All treatments had a common bloom period with a high positive growth rate from days 6 to 10. Consequently, this period was hereafter called the bloom period for all treatments. The period before, from days 0 to 5, was called the pre-bloom period and the period after, from days 11 to 18, the post-bloom period. Those common periods were used further to statistically test the differences between treatments. Statistical differences of Chl *a* concentration between treatments (two-way RM-ANOVA) in the different bloom periods are presented in **Table 2** and **S1 Fig** in **S1 File**. Temperature had a significant effect on Chl *a* concentrations for all three periods (pre-bloom, bloom and post-bloom; **Table 2**), while zooplankton community had only a significant effect during pre-bloom and post-bloom periods. Interactions between temperature and zooplankton community were strong for the three periods.

## Treatments effect on nutrient dynamics

At the beginning of the experiment, the nutrient concentrations were relatively high. Notably, the mean NH$_4^+$ and NO$_3^-$ concentrations at day 0 in all mesocosms were 0.83 ± 0.24 and 1.22 ± 0.40 μmol L$^{-1}$, respectively, while that of NO$_2^-$ was 0.07 ± 0.011 μmol L$^{-1}$ (**Fig 3**). The

**Table 2. Statistical results of comparison tests for Chl *a* concentration difference between treatments on two factors (Temperature x Zooplankton community [Zoo. Comm.]), for Pre-bloom, bloom and Post-bloom periods.** Bold values indicate significant tests (*p-values* < 0.05).

| | | Pre-bloom | | | | Bloom | | | | Post-bloom | | | |
|---|---|---|---|---|---|---|---|---|---|---|---|---|---|
| | Factor | Test | df | Test value | *p-value* | Test | df | Test value | *p-value* | Test | df | Test value | *p-value* |
| Chl *a* | Temperature | RM-ANOVA | 39 | **6.98** | **0.0118** | RM-ANOVA | 32 | **14.37** | **0.0006** | RM-ANOVA | 53 | **43.55** | **< 0.0001** |
| | Zoo. Comm. | | | **9.59** | **0.0036** | | | 0.10 | 0.7509 | | | **48.14** | **< 0.0001** |
| | Interact. | | | **5.51** | **0.024** | | | **68.35** | **< 0.0001** | | | **210.9** | **< 0.0001** |

mean $SiO_2$ and $PO_4^{3-}$ concentrations were 3.99 ± 0.80 and 0.84 ± 0.14 μmol L$^{-1}$, respectively. Nutrients showed three general trends in the control (**Fig 3**): 1) $NH_4^+$ concentrations on day 3 showed a general decrease until the end of the bloom, followed by almost stable concentrations until the end of the experiment, 2) $NO_2^-$ and $NO_3^-$ showed stable concentrations until the middle of the experiment, followed by a decrease; and 3) $SiO_2$ and $PO_4^{3-}$ concentrations showed an initial decrease in concentration after a quick pulse on day 2, followed by an increase in the middle of the experiment, and finally a decrease. The non-control treatments (T, MicroZ and TMicroZ) showed similar trends to the control, with several exceptions described hereafter. In T treatment, $NH_4^+$ concentrations showed a lower decrease, and had generally higher concentrations than those in the control (**Fig 3A**). All non-control treatments showed generally higher $NO_2^-$ and $NO_3^-$ concentrations from the middle of the experiment compared to those in the control (**Fig 3B–3D**), and lower $PO_4^{3-}$ concentrations in the first part of the experiment until day 9 (**Fig 3E**). Statistical differences of nutrient concentrations between treatments (RM-ANOVA) in the different bloom periods identified are presented in **S2 Fig in S1 File** and **S2 Table in S1 File**.

## Treatments effect on taxonomic pigments dynamics, and dominant phytoplankton taxa

The taxonomic pigment dynamics showed five different trends (**Fig 4**): (1) 19HF (Prymnesiophyceae) showed a pattern similar to that of Chl *a*, with an initial increase attaining a maximum concentration on day 9 (for T and MicroZ) or 10 (for control and TMicroZ), followed by a decrease (**Fig 4A**). (2) Fucoxanthin (diatoms) showed a similar trend to Chl *a* but with a second increase at the end of the experiment except for control (**Fig 4B**). (3) Peridinin (dinoflagellates) and Zeaxanthin (cyanobacteria) exhibited maximum peak concentrations during the Chl *a* post-bloom period (**Fig 4C and 4D**). (4) Chlorophyll *b* (green flagellates) and Prasinoxanthin (prasinophytes) had two peaks in concentration: the first peak during the pre-bloom period occurring in the heated treatments (T and TMicroZ), and a second during the bloom period in all treatments (**Fig 4E and 4F**). (5) Alloxanthin concentrations decreased throughout the experiment (**Fig 4G**), especially in the control. Statistical differences of pigment concentration between treatments (RM-ANOVA) in the different bloom periods are presented in **S3 Fig in S1 File** and **S3 Table in S1 File**.

Microscopic identification revealed that at the beginning of the experiment, phytoplankton communities >5 μm cell diameter were dominated, in terms of abundance, by small phytoplankton taxa such as Cryptophyceae (*Teleaulax acuta;* **S4 Table in S1 File**), undifferentiated Prymnesophyceae and undifferentiated Chlorophyceae in all treatments. Those taxa remained dominant until the end of the main bloom (day 0, 3 and 9). During the whole experiment, diatoms were dominated by *Chaetoceros* spp. and *Cyclotella* spp. except the last day (day 18) where they were dominated by multiple taxa such as the latter and *Guinardia* sp., *Pseudo-*

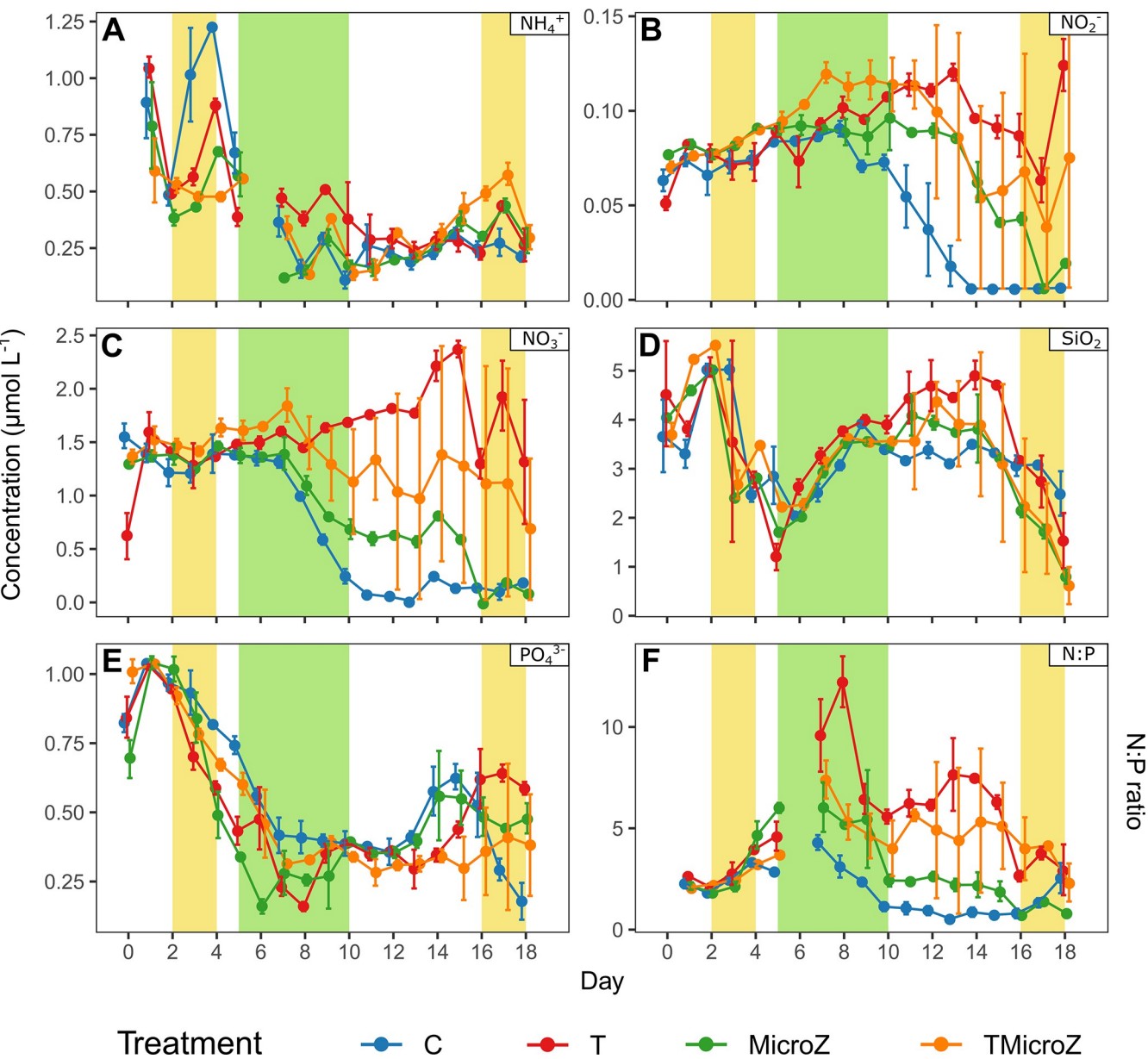

**Fig 3. The daily mean (±range of the observations) of nutrient concentrations in the different treatments over the course of the experiment.** (A) $NH_4^+$, (B) $NO_2^-$, (C) $NO_3^-$, (D) $SiO_2$, (E) $PO_4^{3-}$, and (F) N:P ratio (with N = $NH_4^+$ + $NO_2^-$ + $NO_3^-$). C, control; T, water heated +3˚C; MicroZ, mesozooplankton exclusion; TMicroZ, water heated at +3˚C with mesozooplankton exclusion. Green background indicates the main bloom period common to all treatments and yellow background indicates the early and late blooms identified in T and TMicroZ treatments.

*nitzschia* spp. and *Licmophora* sp., depending on the treatment. During the post-bloom (day 14), *Gymnodinium* spp. dominated in T, MicroZ, and TMicroZ treatments, but not in C.

## Treatments effect on Pico- and Nanophytoplankton dynamics

Phytoplankton abundances showed contrasting trends between groups (**Fig 5**). Mean Cyanobacteria abundance (Cyano) in all treatments at day 0 started with 550 ± 122 cells $mL^{-1}$ and decreased thereafter and reached stability on day 6 until the end of the experiment (control, MicroZ and TMicroZ) (**Fig 5A**). Only in the T treatment, the Cyano abundance increased on

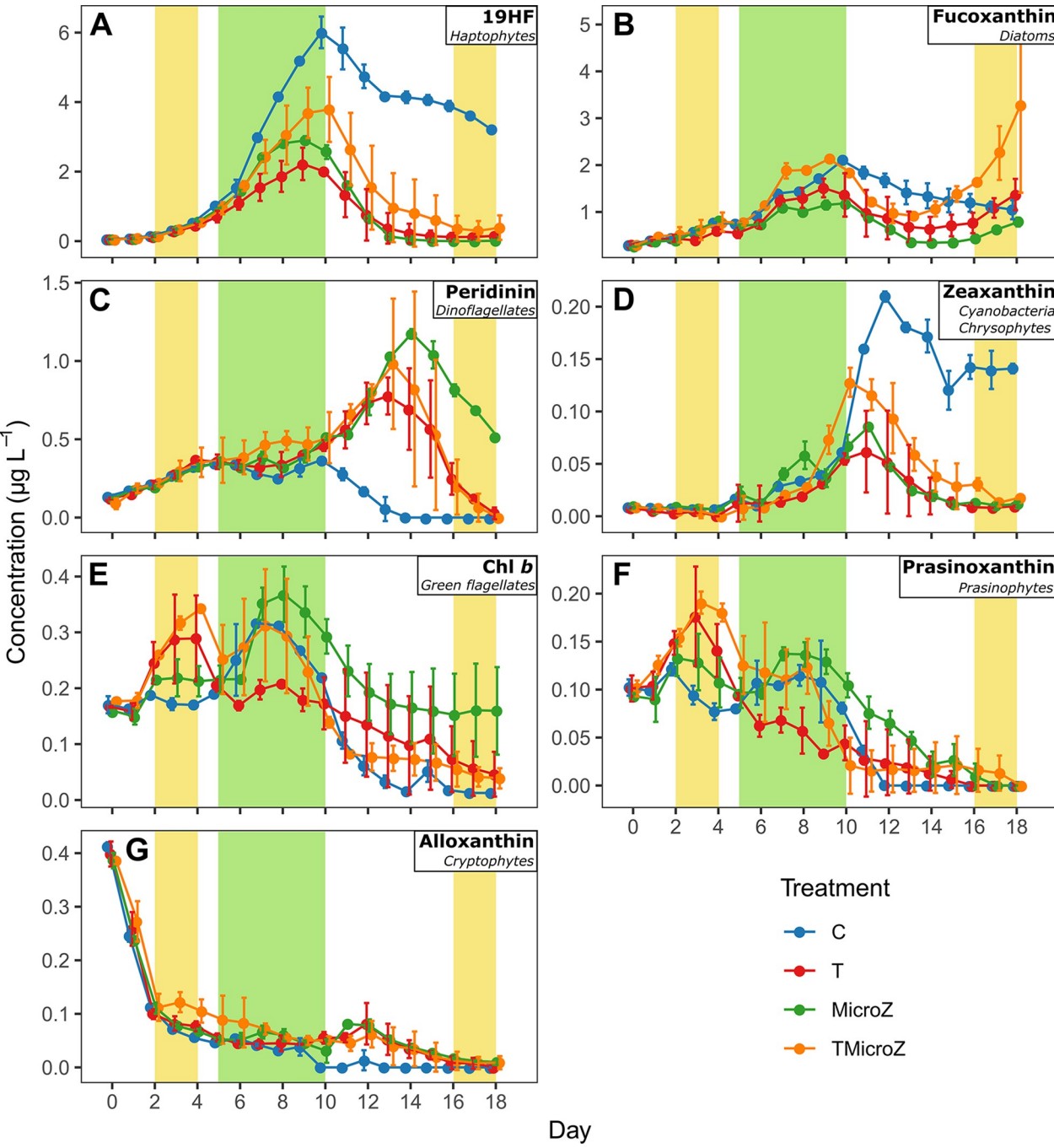

**Fig 4. Mean pigment concentrations (±range of the observations) in the different treatments.** (A) 19HF, (B) Fucoxanthin, (C) Peridinin, (D) Zeaxanthin, (E) Chl *b*, (F) Prasinoxanthin, and (G) Alloxanthin. C, control; T, water heated +3˚C treatment; MicroZ, mesozooplankton exclusion treatment; TMicroZ, water heated +3˚C and mesozooplankton exclusion. Green background indicates the main bloom period common to all treatments and yellow background indicates the early and late blooms identified in T and TMicroZ treatments.

day 6 and remained stable until the end of the experiment. However, the range of the observations for the T treatment was wide, as a pronounced Cyano abundance was only observed in one out of the two mesocosms (T2). The mean Picophytoeukaryote (Picoφ) abundance in all treatments was $1.68 \times 10^4$ cell $mL^{-1}$ on day 0, and thereafter increased, reaching a peak on day

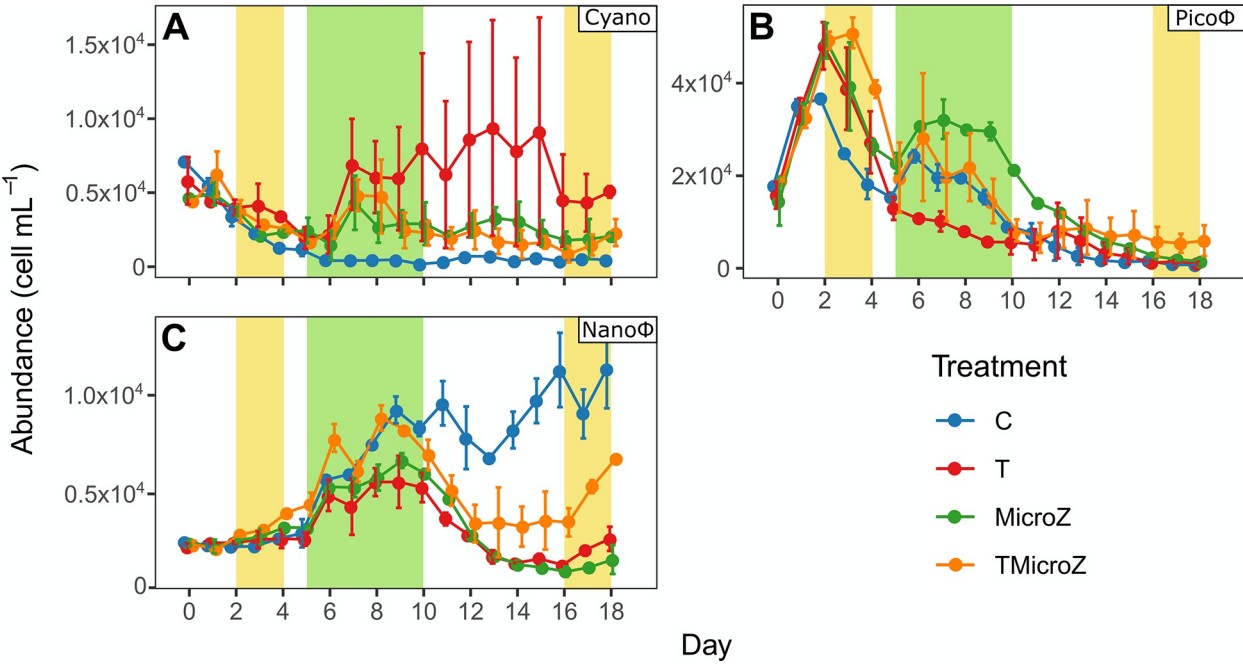

**Fig 5. Mean phytoplankton abundances (±range of the observations) in the different treatments.** (A) Cyanobacteria (Cyano), (B) Picophytoeukaryotes (Picoφ), and (C) Nanophytoeukaryotes (Nanoφ). C, control; T, water heated +3˚C treatment; MicroZ, mesozooplankton exclusion treatment; TMicroZ, water heated +3˚C and mesozooplankton exclusion treatment. Green background indicates the main bloom period common to all treatments and yellow background indicates the early and late blooms identified in T and TMicroZ treatments.

2, which was delayed to day 3 in the TMicroZ treatment (**Fig 5B**). Its abundance decreased and then formed a second but lower peak during the main bloom period, in all treatments except the T treatment. The abundance decreased to a minimum value of $0.24 \pm 0.23 \times 10^4$ cells mL$^{-1}$ on day 18 in all treatments. The mean Nanophytoeukaryote (Nanoφ) abundance in all treatments was $0.24 \pm 0.01 \times 10^4$ cells mL$^{-1}$ at the beginning of the experiment, which increased from day 5 to a maximum during the main bloom period on day 8 or 9 depending on the treatment (**Fig 5C**). In the T, MicroZ, and TMicroZ treatments, it decreased thereafter to reach a plateau during the post-bloom period before a final increase, especially in the TMicroZ treatment, started on day 16 until the end of the experiment, during the late-bloom period. In contrast, for the control, Nanoφ abundance keeps increasing to reach a maximum value on day 18 during the post-bloom period. Statistical differences of pico-and nanophytoplankton abundances between treatments (RM-ANOVA) in the different bloom periods are presented in **S4 Fig in S1 File** and **S5 Table in S1 File**.

## Treatments effect on microzooplankton dynamics, and mesozooplankton composition

Microprotozooplankton abundance started with ca. $2.00 \pm 0.14 \times 10^4$ ind L$^{-1}$ at day 0 in C, T and TMicroZ with slightly higher initial abundances of around $2.78 \pm 0.14 \times 10^4$ ind L$^{-1}$ in MicroZ (**Fig 6A**). Abundances decreased in all treatments until day 7 but slower in MicroZ, resulting in a significantly higher abundance during the bloom period in the latter compared to the other (**S5 Fig in S1 File**). Thereafter, abundances increased slightly until day 13 in T, MicroZ and TMicroZ until a final decrease. The C treatment, however, showed a continuous decline until the end of the experiment. Micrometazooplankton abundances started with $47.50 \pm 38.45$ ind L$^{-1}$ at the beginning of the experiment (**Fig 6B**) but formed intense blooms

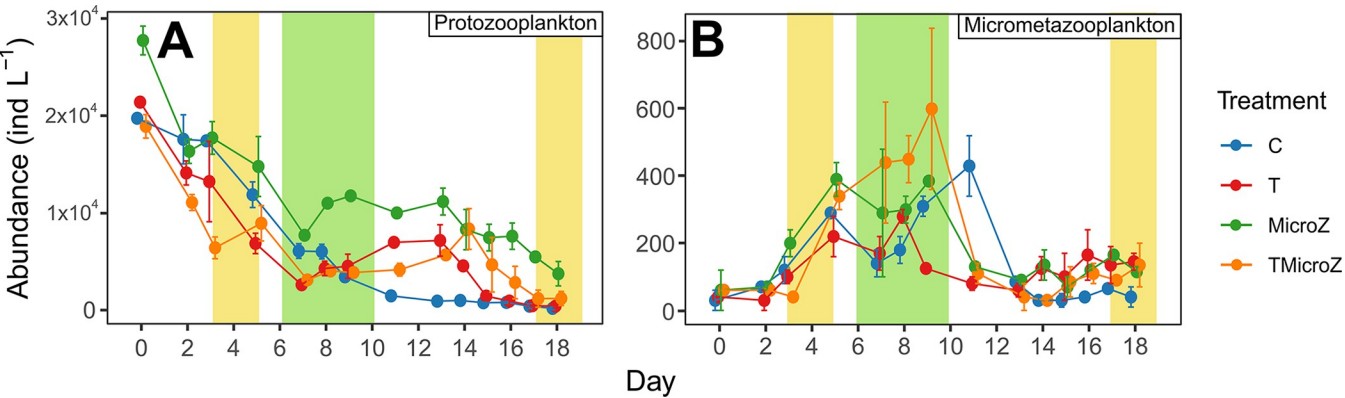

**Fig 6. Mean microzooplankton abundances (±range of the observations) in the different treatments.** (A) Microprotozooplankton, and (B) Micrometazooplankton. C, control; T, water heated +3˚C treatment; MicroZ, mesozooplankton exclusion treatment; TMicroZ, water heated +3˚C and mesozooplankton exclusion treatment. Green background indicates the main bloom period common to all treatments and yellow background indicates the early and late blooms identified in T and TMicroZ treatments.

(up to 6- to 10-fold depending on the treatments). During the bloom, micrometazooplankton abundances were significantly higher in TMicroZ than in C and T (**S5 Fig in S1 File**). After the bloom, their abundances decreased to reach a plateau around 91.86 ± 55.95 from day 13 until the end of the experiment. Micrometazooplankton was dominated by rotifers but also comprised of D-stage veliger larvae, copepod nauplii and polychaete larvae. Temperature had a significant effect on microprotozooplankton abundances during pre-bloom and bloom periods (**Table 3**; *p-values* < 0.001), and on micrometazooplankton abundances during the pre-bloom period only. The initial zooplankton community structure had a significant effect on the microprotozooplankton abundances during the bloom and post-bloom periods (**Table 3**; *p-values* < 0.001), and on microzooplankton abundances during pre-bloom and bloom periods (*p-values* < 0.05). Interaction between temperature and the initial zooplankton community had a significant effect on microprotozooplankton abundances during pre-bloom and bloom periods (*p-values* < 0.05; not applicable for post-bloom), but not for micrometazooplankton abundance.

For mesozooplankton on day 0, the 200-μm screening reduced the abundance of *Oithona* sp. and Polychaeta (MicroZ and TMicroZ; **S6 Table in S1 File**) compared to the 1000-μm screened treatments (control and T). During the phytoplankton bloom (day 7), the mesozooplankton community was dominated by *Acartia* sp. in all treatments. In heated treatments, *Acartia* sp. abundances (T and TMicroZ) were lower than in non-heated treatments (control

**Table 3. Statistical results of comparison tests for Microzooplankton (Microprotozooplankton and Micrometazooplankton) concentration differences between treatments on two factors (Temperature x Zooplankton community [Zoo. Comm.]), for Pre-bloom, bloom and Post-bloom periods.** Bold values indicate significant tests (*p-values* < 0.05).

| | | Pre-bloom | | | | Bloom | | | | Post-bloom | | | |
|---|---|---|---|---|---|---|---|---|---|---|---|---|---|
| | Factor | Test | df | Test value | *p-value* | Test | df | Test value | *p-value* | Test | df | Test value | *p-value* |
| Microprotozoo. | Temperature | RM-ANOVA | 25 | **26.48** | **<0.0001** | RM-ANOVA | 18 | **52.92** | **<0.0001** | Kruskall-Wallis | 1 | 0.14 | 0.7062 |
| | Zoo. Comm. | | | 0.00134 | 0.9711 | | | **20.15** | **0.0003** | | | **18.15** | **<0.0001** |
| | Interact | | | **6.08** | **0.0209** | | | **22.46** | **0.0002** | | | NA | NA |
| Micrometazoo. | Temperature | RM-ANOVA | 25 | **5.41** | **0.0285** | RM-ANOVA | 18 | 1.86 | 0.1897 | RM-ANOVA | 46 | 0.2 | 0.6567 |
| | Zoo. Comm. | | | **4.79** | **0.0382** | | | **13.93** | **0.0015** | | | 0.12 | 0.7294 |
| | Interact | | | 0.47 | 0.5003 | | | 2.85 | 0.1085 | | | 1.09 | 0.3019 |

and MicroZ) on day 7. Oncaeidae were absent in the T, MicroZ, and TMicroZ treatments on day 7, but present in the control. On the contrary, Oncaeidae were more abundant in MicroZ, T, and TMicroZ treatments on day 14, while they were absent in the control.

## Phytoplankton community succession

Phytoplankton community succession and their explanatory factors showed different patterns between treatments (**Fig 7**). Community composition of the non-heated treatments (**Fig 7A** and **7C**) were well explained by RDA1 and RDA2 (C: 94%, and MicroZ: 88%) but less for the heated ones (**Fig 7B** and **7D**; T: 76%, and TMicroZ: 73%). For non-heated treatments (C and MicroZ: **Fig 7A** and **7C**), phytoplankton communities were well differentiated between periods (pre-bloom, bloom and post-bloom) and did not overlap. Transition in communities from a period to another was sharp with no 'in between' communities. On the other hand, heated treatments presented more spread-out communities, highlighted by overlapping ellipses. Transition in communities between two periods was more progressive.

Explanatory variables ordination and correlation tests associated revealed that nutrients, especially nitrogen, strongly altered phytoplankton community composition in the non-heated treatments (**Fig 7A** and **7C**), while it was less the case for communities in the heated treatments (**Fig 7B** and **7D**). Regarding biotic explanatory variables, phytoplankton community composition was mainly influenced by microprotozooplankton in the C and T treatments (**Fig 7A** and **7B**), while in MicroZ and TMicroZ, it was mainly affected by micrometazooplankton (**Fig 7C** and **7D**).

Statistical analysis on phytoplankton beta-diversity index (**Table 4**) revealed that communities were similar between treatments during the pre-bloom period. During bloom period, phytoplankton community composition was significantly different between all treatments, except between C and TMicroZ. During post-bloom period, phytoplankton community composition was significantly different between all treatments. C treatment shared the most important differences with all other treatments (Pseudo-F > 24; $p$-values < 0.001), while T, MicroZ and TMicroZ communities were more similar (Pseudo-F < 10; $p$-values < 0.05), albeit significantly different.

## Discussion

The mesocosm experiment was planned to study how temperature and trophic structure influence the phytoplankton spring bloom that naturally occurs in the Thau lagoon in early April every year [11]. As expected, a main phytoplankton bloom was observed in all the mesocosms a few days after its start, as already reported from previous mesocosm experiments at the same study site in spring [24]. Phytoplankton bloom initiation and dynamics in the mesocosms could be triggered by various factors since both bottom-up and top-down control mechanisms are known to influence bloom dynamics and composition. Notably, during springtime, the trophic cascade structure from overwintering zooplankton plays an important role in the subsequent bloom dynamics and characteristics [9,18,46,47].

### The role of mesozooplankton and microzooplankton in the Thau Lagoon trophic cascade

Mesozooplankton removal increased microzooplankton abundance in ambient conditions (C vs. MicroZ) and triggered a strong reduction of the bloom amplitude. Microzooplankton (both micrometazooplankton and microprotozooplankton) was found to be the main consumer of phytoplankton in the present study. This result is not surprising as natural phytoplankton communities in the Thau Lagoon are mainly characterized by small-sized species

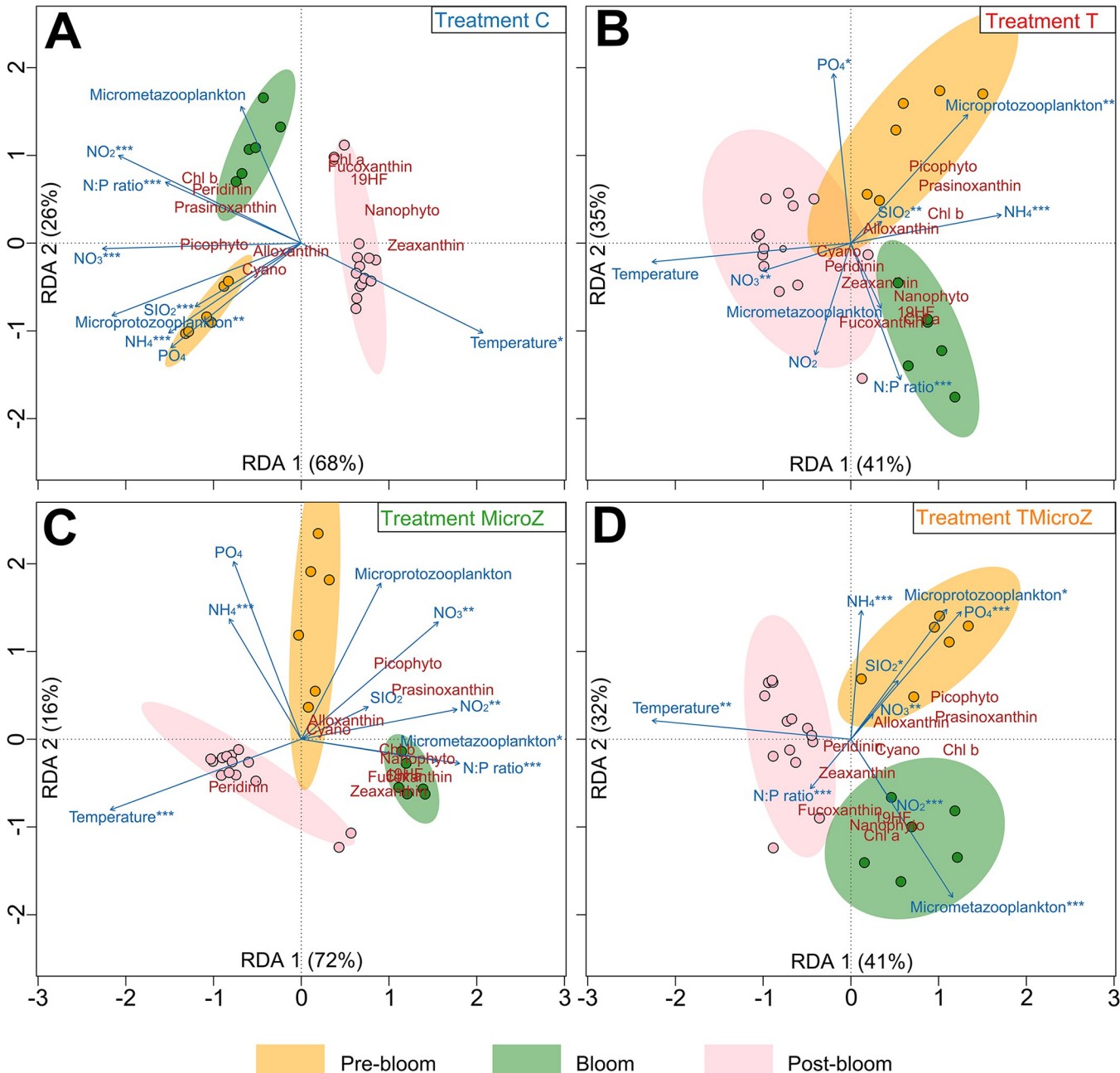

**Fig 7. Redundancy Analysis (RDA) between phytoplankton composition (pigment and cytometry) and environmental parameters.** (A) control (C) treatment, (B) heated +3°C (T) treatment, (C) the mesozooplankton exclusion treatment (MicroZ treatment), and (D) the heated +3°C and mesozooplankton exclusion treatment (TMicroZ). Dots present sampling dates community composition. Blue arrows present the projections of environmental parameters. Ellipses represents 95% confidence distribution interval of community for different periods (Pre-bloom, Bloom and Post-bloom). Significance level of ANOVA: * = $p$-value < 0.05; ** = $p$-value < 0.01; *** = $p$-value < 0.001. Red labels present projection of the pigments and cytometry groups. The proximity between environmental parameters or pigments and cytometry groups and sampling dates indicates characteristic associations.

that are prone to micrograzer predation [11,48,49]. Even diatoms, that are usually considered as of larger cell size in a variety of environments [50], are rarely >15 μm diameter in the Thau Lagoon [49]. This pattern was not different in the present experiment, as *Chaetoceros* spp. was the dominant species throughout the bloom and within this size range. As stressed by previous

**Table 4. PERMANOVA tests on phytoplankton community composition differences between treatment, on pre-bloom, bloom, and post-bloom periods.**

| | Pre-bloom | | Bloom | | Post-bloom | |
|---|---|---|---|---|---|---|
| | Pseudo-F | *p*-value | Pseudo-F | *p*-value | Pseudo-F | *p*-value |
| C vs T | 0.6303 | 0.437 | **18.738** | **0.001 ***** | **33.966** | **0.001 ***** |
| C vs MicroZ | 1.4091 | 0.525 | **17.337** | **0.001 ***** | **38.019** | **0.001 ***** |
| C vs TMicroZ | 3.6882 | 0.063 | 0.5094 | 0.556 | **24.168** | **0.001 ***** |
| T vs MicroZ | 0.1948 | 0.775 | **63.871** | **0.001 ***** | **3.7088** | **0.044 *** |
| T vs TMicroZ | 1.0719 | 0.317 | **9.8251** | **0.001 ***** | **9.4543** | **0.001***** |
| MicroZ vs TMicroZ | 0.4567 | 0.557 | **9.759** | **0.007 **** | **4.3792** | **0.028*** |

Pseudo-F represents the test value. Significance level of PERMANOVA: * = *p*-value < 0.05; ** = *p*-value < 0.01; *** = *p*-value < 0.001. Bold values depicted significant differences in phytoplankton community composition between treatments.

studies [51,52], microzooplankton are considered as principal grazers of autotroph production and major phytoplankton bloom controllers. This is in good accordance with our findings showing that diatoms (fucoxanthin) and prymnesiophytes (19HF) biomass experienced the strongest decrease among all pigments in MicroZ thus highlighting the role of microzooplankton as the principal grazers of phytoplankton. In contrast, in the absence of mesozooplankton (MicroZ), the abundance of both micrometazooplankton and microprotozooplankton was higher than in C during the bloom. Thus, in Thau Lagoon during springtime, mesozooplankton played a minor role as phytoplankton grazers and can thus be mainly considered as secondary consumers of phytoplankton relative to microzooplankton. This trophic structure might, however, vary depending on the phytoplankton composition. The 200 μm screening did not completely remove all mesozooplankton, probably due to the fact that, depending on the taxa's specific dimensions, some organisms are very slender (rather long but thin, as for example *Acartia* sp.) so that they can still pass the mesh. Despite the fact that some few mesozooplankton taxa obviously still passed the mesh during the filling process, the significant difference in bloom amplitude and composition observed in MicroZ compared to C confirmed that the modification of the trophic cascade structure we aimed for was successful.

## Temperature and trophic cascade structure conditioned the bloom amplitude: The role of top-down forces under warming

Top-down control by meso- and microzooplankton grazers can affect the phenology, bloom amplitudes and community composition of phytoplankton [18,53]. However, the underlying mechanisms of predator-prey relationships and the distinct roles of specific grazer groups under warmer conditions are not entirely clear yet. In the present study, the planktonic trophic cascade was manipulated by removing mesozooplanktonic predators (> 200 μm) and subjected to warmer conditions. As expected, warming affected natural plankton communities (T treatment) strongly by reducing the phytoplankton spring bloom amplitude significantly due to a strengthening of top-down control. This huge reduction in the bloom amplitude was already reported for the same mesocosm experiment [20,21] based on sensor measurements. However, the role of top-down control was not addressed in the previous study since the focus was only on the C and T treatments. Nonetheless, the increase of grazing rates reported in [21] under warming are in accordance with our results suggesting that the phytoplankton suppression in T was due to an increase in both mesozooplankton and microzooplankton predation pressure, affecting the whole phytoplankton community. Similar results were found in previous mesocosm studies in other environments [13,54–56], highlighting the potential role of predators in bloom control under warmer conditions, rather than an acceleration in nutrient

depletion. This specific trophic cascade under warmer condition was also observed *in situ* in the Thau Lagoon and hypothesized as the main process causing a weak spring phytoplankton bloom during an abnormal warm year [11,49]. This suggests that the four treatments described in the present paper capture well the main ecological processes within the plankton community in the Thau Lagoon over different time scales and in relation to changes in biotic and abiotic conditions.

The general increase in the predator pressure under warming is a well-known process that can be explained by the differential response between autotrophy and heterotrophy toward the metabolic theory of ecology [17,57,58]. Temperature affects metabolic processes in heterotrophs, notably growth and uptake rates, more than in autotrophs [17,58]. While warmer conditions accelerate the metabolic rates of autotrophs to some extent, this cannot compensate for the increased grazing pressure of predators. This differential response disrupts predator-prey interactions with warming, thus affecting the growth-grazing balance under warmer conditions depending on the thermal responses of different consumer levels.

One major result of the present study is that when removing predators > 200 μm (MicroZ and TMicroZ) the bloom amplitude showed an opposite pattern than C and T in response to warming. In fact, when mesozooplankton > 200 μm was initially removed from the community, the heated treatment (TMicroZ) showed a higher phytoplankton biomass during the bloom than in the non-heated treatment (MicroZ). Some previous studies also reported an increase in peak amplitudes of phytoplankton under warmer conditions [24,59]. In these studies, an enhanced phytoplankton production was either due to a modification of the trophic cascade that reduced the abundance of the primary consumers, or because of the low initial abundance of planktonic predators. The increase in phytoplankton biomass with warming in TMicroZ relative to MicroZ during the present study supports these findings. Without their natural predators (mesozooplankton) and with enhanced grazing rates under warmer conditions [20], micrometazooplankton that serves as secondary consumers were favored (e.g., rotifers and polychaete). This, in turn, reduced to some extent the abundance of microprotozooplankton (e.g., naked ciliates and tintinnids) as another predator of phytoplankton. This allowed phytoplankton to benefit from warming and to develop freely in the absence of predation. This showed that microzooplankton (both micrometazooplankton and microprotozooplankton) plays a significant role in controlling phytoplankton biomass. Although, more complex interspecific predator-prey interactions might have occurred within the microzooplankton, and their distinct role is still a 'blackbox' that needs further investigation. The present study highlighted that the initial predator community composition and the trophic cascade structure strongly drive the phytoplankton response to warming and thus the amplitude of the bloom. Depending on the trophic cascade structure, the zooplankton composition and the overwintering conditions, the effect of warmer conditions on spring bloom amplitude from a year to another can vary in a single area, even showing opposite responses as schematically illustrated in Fig 8.

During spring, when nutrients are available in access and light conditions improve, bottom-up forcing is considered to play a pivotal role for phytoplankton bloom initiation [47]. Some studies reported that increases in temperature can cause an increase in nutrient limitation due to faster metabolic rates and uptake by phytoplankton, inducing a faster nutrient depletion [10]. In our study, nutrient concentrations were generally higher at elevated temperatures compared to ambient conditions, and less significant as drivers of the phytoplankton community composition, especially during the bloom and the post-bloom periods. The possibility of transient nutrient-limited conditions under warming cannot be excluded at specific days notably for phosphorous during the bloom in the T treatment (on day 8), but was not a major driver of the bloom amplitude and composition. In the present study, warming

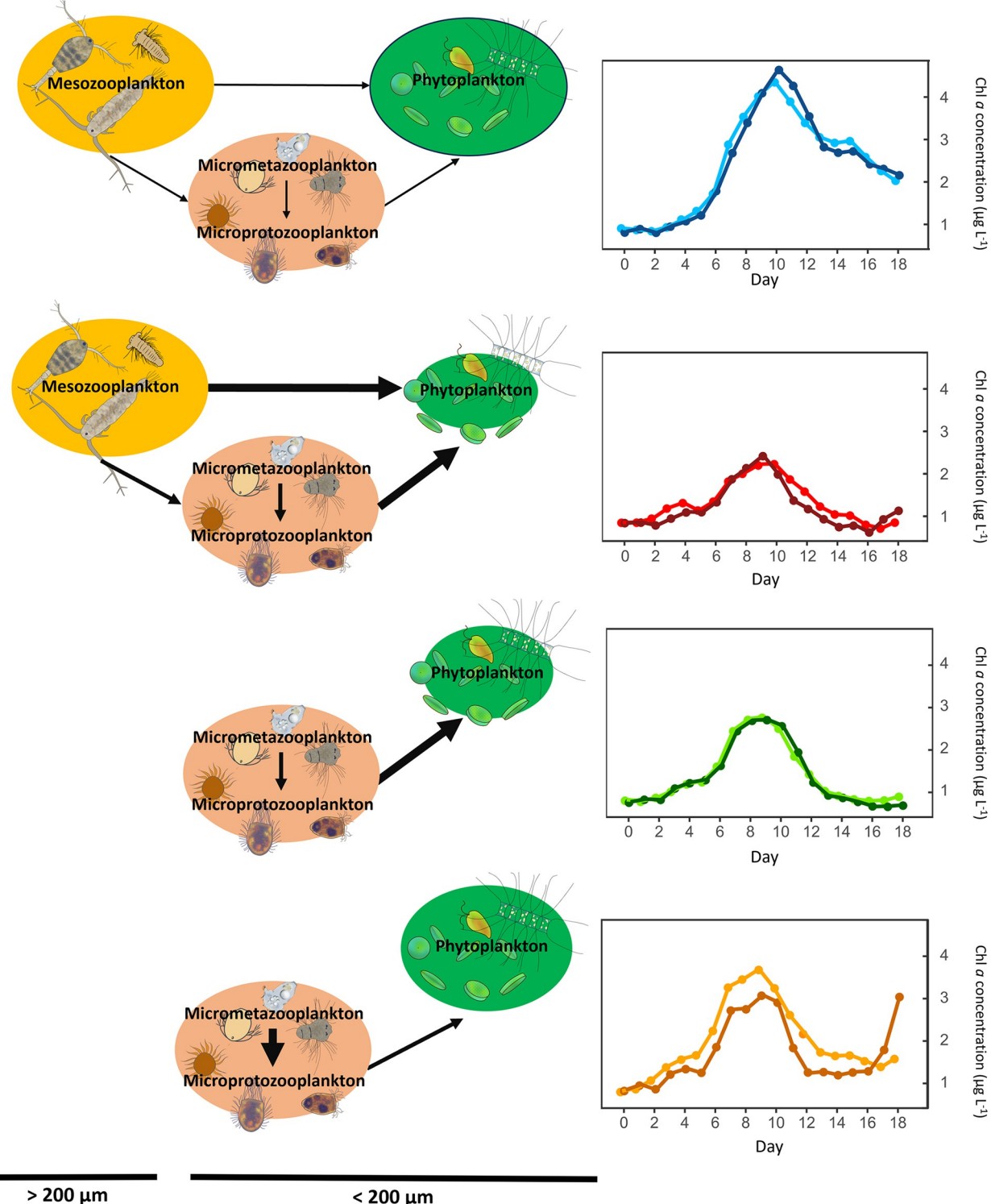

**Fig 8. Treatments effect on the trophic cascade and the bloom amplitude.** Mesozooplankton and microzooplankton compartments represent the initial trophic cascade structure presents at the beginning of the experiment. The size of the arrow represents the hypothesized strength of the grazing pressure (from both metabolism effects of warming and abundance effect). Arrows between micrometazooplankton and microporotozooplankton are hypothesized predator-prey interaction. The size of the phytoplankton compartment represents the maximum amplitude of the bloom observed.

enhanced more the top-down control of phytoplankton bloom amplitude and termination rather than bottom-up processes that could have played a role only occasionally.

## Warming and trophic cascade effect on phytoplankton succession and composition

In the present experiment, warming induced an early bloom of small green algae and the main phytoplankton bloom was one day shorter than under ambient conditions (C). Such a faster bloom onset and termination were already observed or modeled in various systems [47,55,60–62]. However, it should be stressed that our observation was based on only one time point during our mesocosm study and thus doesn't necessarily apply to larger scales. It was often suggested that this was due to the acceleration of phytoplankton metabolism with warming under light saturated conditions [63,64] such as those experienced in the present study, that could have triggered this early bloom of small green algae. Because of their favorable size to volume ratio, small green algae might have benefited from warming due to an acceleration of nutrient uptake [65,66]. However, as growth and grazing of heterotroph organisms, especially microzooplankton, was also accelerated under warming, this led to alterations of the phytoplankton-grazers dynamic that triggered this early bloom termination. This acceleration might also have induced the late bloom of diatoms observed at elevated temperatures during the last two days of the experiment. Overall, the present study shows that warming can accelerate predator-prey interactions and induce shifts in plankton dynamics in this shallow coastal lagoon.

Besides the observed shifts in phytoplankton timing, warming also modified the phytoplankton composition during bloom and post-bloom periods. During the bloom, that was dominated by Nanoφ, diatoms (fucoxanthin; mainly *Chaetoceros spp.*) and prymnesiophytes (19HF) at ambient temperature conditions, warming with zooplankton < 1000 μm (T) lead to a dominance shift towards dinoflagellates (peridinin; mainly *Gymnodinium* sp.), cyanobacteria (Cyano and zeaxanthin) and cryptophytes (alloxanthin). This finding was already reported from the same experiment [21]. However, in the present study, modifying the trophic cascade structure revealed that this change of phytoplankton composition was mainly due to top-down forces. Removing the mesozooplankton (C vs MicroZ) enhanced the dominance of small phytoplankton (Picoφ, small green algae (Chl *b*) and prasinophytes (prasinoxanthin)), while under warming (TMicroZ) it was a combination between the two effects, promoting both small phytoplankton and other minor groups (dinoflagellates, cyanobacteria and cryptophytes). The present findings corroborate previous studies indicating significant changes in phytoplankton bloom communities [55], often attributed to predator influence [66,67]. However, this study emphasizes the complex interplay of trophic cascade structure and warming in this process, which can deeply reshape bloom composition depending on the initial zooplankton and phytoplankton composition. In all non-control treatments, this led to a striking shift of post-bloom phytoplankton community composition compared to C, generally dominated by dinoflagellates, cryptophytes and small green algae. *In situ*, this drastic shift in phytoplankton composition influenced by warming, trophic cascade structure and the interplay between both these factors could cause substantial alterations in food web structure and functioning. Promoting small phytoplankton and dinoflagellates might strengthen microbial food webs *per se* and the role of microzooplankton in energy transfer up the food web. On the other hand, one should note that phytoplankton species harbor a substantial diversity of pigments and their concentration and proportion changes in relation to environmental parameters (e.g., light intensity). Therefore, the pigment concentrations/proportions do not necessarily reflect the absolute real community diversity.

In conclusion, even if forecasts on ecosystems response to global warming based on results of short-term experiments must be considered with caution, mesocosm experiments can be useful tools to highlight complex mechanistic processes that can occur at the lower planktonic food web level under warming. The present study highlighted the tight coupling between warming and trophic cascades that can impact phytoplankton bloom amplitude, succession, and composition. Eventually, this suggests that the responses of phytoplankton spring blooms to short- or long-term warming strongly depends on biological interactions, notably zooplankton community composition and trophic cascade structure in shallow coastal zones. Therefore, to better assess future trends in phytoplankton blooms in the context of warming, it is crucial to take zooplankton community composition and food web structure into account, such as overwintering zooplankton, micro-mesozooplankton interactions, zooplankton grazing and direct effects of warming on zooplankton metabolism.

## Supporting information

**S1 File. S1 Fig.** Median Chl $a$ concentrations in the different treatments during pre-bloom (A), bloom (B), and post-bloom periods (C). Blue indicates the control treatment (C), red, green, and yellow indicated the heated (T), mesozooplankton exclusion (MicroZ), and heated and mesozooplankton exclusion (TMicroZ) treatments, respectively. Significance level of RM-ANOVAs: * = $p$-value < 0.05; ** = $p$-value < 0.01; *** = $p$-value < 0.001. The letters indicate significant differences between treatments based on post hoc pairwise tests. Boxplots that share the same letter are not significantly different. Boxplots with different letters differ are significantly different ($p$-value < 0.05). Nonsignificant RM-ANOVAs has no stars or letters. **S2 Fig.** Median nutrient concentrations in the mesocosms between the different treatments for pre-bloom, bloom, and post-bloom periods. Nutrients are $NH_4^+$ (A, B, and C), $NO_2^-$ (D, E, and F), $NO_3^-$ (G, H, and I), $SiO_2$ (J, K, and L), and $PO_4^{3-}$ (M, N and O). Blue, control treatment (C); red, green, and yellow, the heated (T), mesozooplankton exclusion (MicroZ), and heated and mesozooplankton exclusion (TMicroZ) treatments, respectively. Significance level of RM-ANOVAs: * = $p$-value < 0.05; ** = $p$-value < 0.01; *** = $p$-value < 0.001. The letters indicate significant differences between treatments based on post hoc pairwise tests. Boxplots that share the same letter are not significantly different. When the letters differ, they are significantly different ($p$-value < 0.05). A lack of asterisks and letters indicates RM-ANOVAs was nonsignificant. **S3 Fig.** Median concentrations of taxonomic pigments in the different treatments during pre-bloom, bloom, and post-bloom periods: Chl $b$ (A, B, and C), Prasinoxanthin (D, E, and F), Zeaxanthin (G, H, and I), Alloxanthin (J, K, and L), 19HF (M, N, and O), Fucoxanthin (P, Q, and R), and Peridinin (S, T, and U). Blue, control (C); red, green, and yellow, heated (T), mesozooplankton exclusion (MicroZ), and heated and mesozooplankton exclusion (TMicroZ) treatments, respectively. Significance level of RM-ANOVAs: * = $p$-value < 0.05; ** = $p$-value < 0.01; *** = $p$-value < 0.001. The letters indicate significant differences between treatments based on post-hoc pairwise tests. Boxplots that share the same letter are not significantly different. When the letters differ, they are significantly different ($p$-value < 0.05). Nonsignificant RM-ANOVAs has no stars or letters. **S4 Fig.** Median abundance of phytoplanktonic groups in the different treatments during pre-bloom, bloom, and post-bloom periods: Cyano (A, B, and C), Pico$\varphi$ (D, E, and F), and Nano$\varphi$ (G, H, and I). Blue, control (C); red, green, and yellow, heated (T), mesozooplankton exclusion (MicroZ), and heated and mesozooplankton exclusion (TMicroZ) treatments, respectively. Significance level of RM-ANOVAs: * = $p$-value < 0.05; ** = $p$-value < 0.01; *** = $p$-value < 0.001. The letters indicate significant differences between treatments based on post hoc pairwise tests. Boxplots that share the same letter are not significantly different. When the letters differ, they are significantly

different ($p$-value < 0.05). Nonsignificant RM-ANOVAs has neither stars nor letters. **S5 Fig.** Median abundance of protozooplankton (A, B and C) and micrometazooplankton (D, E and F) in the different treatments during pre-bloom, bloom, and post-bloom periods. Blue, control (C); red, green, and yellow, heated (T), mesozooplankton exclusion (MicroZ), and heated and mesozooplankton exclusion (TMicroZ) treatments, respectively. Significance level of RM-A-NOVAs: * = $p$-value < 0.05; ** = $p$-value < 0.01; *** = $p$-value < 0.001. The letters indicate significant differences between treatments based on post hoc pairwise tests. Boxplots that share the same letter are not significantly different. When the letters differ, they are significantly different ($p$-value < 0.05). Nonsignificant RM-ANOVAs has neither stars nor letters. **S6 Fig.** Redundancy Analysis (RDA) between phytoplankton composition (pigment and cytometry and environmental parameters. (A) Pre-bloom period, (B) Bloom period, and (C) Post-Bloom periods. Dots present sampling dates community composition. Blue arrows present the projections of environmental parameters. Red labels present projection of the pigments and cytometry groups. The proximity between environmental parameters or pigments and cytometry groups and sampling dates indicates characteristic associations. Ellipses represents 95% confidence distribution interval of community for each treatment (C, T, MicroZ and TMicroZ). Ellipses were not represented for Pre-bloom (A) because they were overlapping and communities showed no significant differences. **S1 Table**. Light penetration mean in the mesocosms and diffuse attenuation coefficient (Kd). **S2 Table**. Statistical results for nutrients ($NH_4^+$, $NO_2^-$, $NO_3^-$, Si, $PO_4^{3-}$) concentration differences between treatments on two factors (Temperature x Zooplankton community [Zoo. Comm]), for Pre-Bloom, Bloom and Post-Bloom periods. Bold values indicate significant tests (p-values < 0.05). **S3 Table**. Statistical results for Pigments (Chl $b$, Prasi, Zea, Allo, 19HF, Fuco, Peri) concentration differences between treatments on two factors (Temperature x Zooplankton community [Zoo. Comm]), for Pre-Bloom, Bloom and Post-Bloom periods. Bold values indicate significant tests (p-values < 0.05). **S4 Table.** Dominant phytoplankton mean abundance (±range) identified using Microscopy in each treatment (in cell mL$^{-1}$). **S5 Table**. Statistical results for Pico- and Nano-phytoplankton (Cyano, Pico and Nano) concentration differences between treatments on two factors (Temperature x Zooplankton community [Zoo. Comm]), for Pre-Bloom, Bloom and Post-Bloom periods. Bold values indicate significant tests (p-values < 0.05). **S6 Table.** Dominant zooplankton taxa mean abundance (±range) identified using stereomicroscope in each treatment (in ind L$^{-1}$).
(DOCX)

**S1 Data.**
(XLSX)

## Acknowledgments

We would like to acknowledge Katharina T. Bading for analyzing microzooplankton data and providing precious feedback for this paper. We would like to thank David Parin, Solenn Soriano and Remy Valdès for their precious technical implications in the experimental setup, sampling and analysis. We are grateful to the numerous students who participated in this experiment performing the sampling and analyses and more generally helping to run this experiment. We acknowledge especially Julien Dupont for the HPLC sampling and analysis, and Aurélien Reymond for microscopic phytoplankton enumeration. We thank Maxime Thibault, Ariadna Garcia-Astillero Honrado and Camille Suarez-Bazille for the nutrient analysis.

## Author Contributions

**Conceptualization:** Behzad Mostajir, Francesca Vidussi.

**Data curation:** Thomas Trombetta, Behzad Mostajir, Justine Courboulès, Sébastien Mas, Francesca Vidussi.

**Formal analysis:** Thomas Trombetta.

**Funding acquisition:** Behzad Mostajir, Francesca Vidussi.

**Investigation:** Thomas Trombetta, Behzad Mostajir, Justine Courboulès, Maria Protopapa, Sébastien Mas, Nicole Aberle, Francesca Vidussi.

**Methodology:** Thomas Trombetta, Behzad Mostajir, Francesca Vidussi.

**Supervision:** Behzad Mostajir, Nicole Aberle, Francesca Vidussi.

**Visualization:** Thomas Trombetta.

**Writing – original draft:** Thomas Trombetta.

**Writing – review & editing:** Thomas Trombetta, Behzad Mostajir, Justine Courboulès, Maria Protopapa, Sébastien Mas, Nicole Aberle, Francesca Vidussi.

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
