## [Decision Letter · Decision Letter 0]

15 May 2024

PONE-D-24-12240Warming and trophic structure tightly controls phytoplankton bloom amplitude, composition and succession.PLOS ONE

Dear Dr. Trombetta,

Thank you for submitting your manuscript to PLOS ONE. After careful consideration, we feel that it has merit but does not fully meet PLOS ONE’s publication criteria as it currently stands. Therefore, we invite you to submit a revised version of the manuscript that addresses the points raised during the review process.

I have now received two reviews of your manuscript. While both reviewers expressed appreciation for your study, one reviewer raised significant concerns regarding the clarity of the introduction, as well as your statistical approach.

Specifically, this reviewer highlighted the need for improvement in the description of aims of the study. I concur with this assessment. Given that your study explores bottom-up and top-down effects, along with indirect interactions, it is imperative to provide a detailed explanation on how you differentiate between these concepts and the consequences for your predictions in your specific study. E.g. what are the predicted responses when excluding large zooplankton?

Furthermore, the reviewer pointed out that your experimental design is structured for a two-way ANOVA rather than a one-way ANOVA, which warrants attention.

Additionally, both reviewers offered various other comments, underscoring the importance of addressing each point comprehensively in your response letter. It is crucial that you provide detailed explanations regarding how you have addressed these comments.

We look forward to receiving your revised manuscript.

Kind regards,

Peter Eklöv

Academic Editor

PLOS ONE

Journal Requirements:

'This study was part of the Photo-Phyto project funded by the French National Research Agency (ANR-14-CE02-0018) financing also the PhD of TT. The experiment was opened to transnational access throughout the AQUACOSM project (European Union’s Horizon 2020 research and innovation program H2020/2017-2020 under grant agreement n°731065.) which financed NAM, and MP participation. Microscopy and cytometry equipment were provided by the MICROBEX platform of MARBEC/CeMEB LabEX with the support of LabEx CeMEB, an ANR "Investissements d'avenir" program (ANR‐10‐ LABX‐04‐01)."

3. We noted in your submission details that a portion of your manuscript may have been presented or published elsewhere. [Flow cytometry data for C and T treatments were published in Courboulès et al 2022.

Pigment data for C and T treatments were published in Soulié et al 2022.

Nutrients and Temperature data for C and T treatments were published in Soulié et al 2022 and Courboulès et al 2022.

These inclusions do not constitue dual publication as it was either 1) background data, 2) not associated to data from MicroZ and TMicroZ treatments, 3) presenting the data under a different focus and research question (i.e. the role of zooplankton). Comparison with these studies are discussed in the 'Discussion' part of the manuscript.] Please clarify whether this [conference proceeding or publication] was peer-reviewed and formally published. If this work was previously peer-reviewed and published, in the cover letter please provide the reason that this work does not constitute dual publication and should be included in the current manuscript.

5. Please upload a copy of Supporting Information Figure/Table/etc. Supporting Figures S1-S6 which you refer to in your text on your Supporting Information file.

Reviewers' comments:

Reviewer's Responses to Questions

**Comments to the Author**

1. Is the manuscript technically sound, and do the data support the conclusions?

Reviewer #1: Yes

Reviewer #2: Yes

2. Has the statistical analysis been performed appropriately and rigorously? 

Reviewer #1: Yes

Reviewer #2: No

3. Have the authors made all data underlying the findings in their manuscript fully available?

Reviewer #1: Yes

Reviewer #2: Yes

4. Is the manuscript presented in an intelligible fashion and written in standard English?

Reviewer #1: Yes

Reviewer #2: Yes

5. Review Comments to the Author

Reviewer #1: Warming and trophic structure tightly controls phytoplankton bloom amplitude, composition and succession.

The authors conducted mesocosm experiments in Thau Lagoon in Southern France to explore the effects of warming and zooplankton on the annual spring phytoplankton bloom. They found that temperature and the exclusion of mesozooplankton decreased the amplitude of the spring phytoplankton bloom, while the combination of these two parameters had almost the same bloom amplitude as the control. They concluded that mesozooplankton plays a role in controlling spring bloom dynamics and that their role could increase with warming.

The manuscript is well-written, concise, and understandable. However, throughout the text several typos and grammatical errors appear, which should be detected by any spell-check programme. I hence suggest checking the language apart from the few points that are explained below.

Specific points

Abstract: None, it is well-written.

Introduction:

Line 63: Here you write that experimental studies have shown interspecific differences, but there is only one study referenced. Please add more references.

Material and Methods:

Line 247: It states that after log-transformation, data normality and homogeneity was assumed. Is there a reason for not testing this with the same tests as before transformation?

Results: None.

Discussion: None.

Figures:

Fig. 4: You could consider linking this figure better to Table 1. For example by writing the taxa corresponding to the pigments directly on the figure or in the figure-text.

Supplementary Material: None.

Reviewer #2: The research question, experimental design, and data gathered are of excellent quality. Still, the primary statistical analysis is inappropriate: the authors should perform a two-way RM-ANOVA instead of a one-way RM-ANOVA.

More comments are detailed in the attachment file.

6. PLOS authors have the option to publish the peer review history of their article (what does this mean?). If published, this will include your full peer review and any attached files.

Reviewer #1: No

Reviewer #2: No

---

## [Author Response · Author response to Decision Letter 0]

2 Jul 2024

Dear Academic Editor and Reviewers,

We would like to thank the Academic Editor and the two Reviewers for the constructive comments, suggestions and advices that significantly improved this study and the manuscript. We are grateful for their fruitful contributions.

Major changes were made on:

• Better defining the aim of this study.

• Statistical analysis, by performing two-ways RM-ANOVA with treatments interactions.

Please, find attached the document where we answer point-by-point to the comments made by the Reviewers.

Thank you again for your expertise and time to review this study.

Best wishes,

T. Trombetta, B. Mostajir, J. Courboulès, M. Protopapa, S. Mas, N. Aberle and F. Vidussi

---

## [Decision Letter · Decision Letter 1]

23 Jul 2024

PONE-D-24-12240R1Warming and trophic structure tightly control phytoplankton bloom amplitude, composition and succession.PLOS ONE

Dear Dr. Trombetta,

Thank you for submitting your manuscript to PLOS ONE. Your manuscript is essentially accepted pending a very minor revision. One reviewer pointed out that there are no references in the statistical treatment section and suggested you should add some references. I also noticed you did not state what software was used to carry out the statistical analysis, which should be stated for the benefit of the reader.   Therefore, we invite you to submit a revised version of the manuscript that addresses these two suggestions.

We look forward to receiving your revised manuscript.

Kind regards,

Hans G. Dam, Ph. D.

Academic Editor

PLOS ONE

Journal Requirements:

Reviewers' comments:

Reviewer's Responses to Questions

**Comments to the Author**

1. If the authors have adequately addressed your comments raised in a previous round of review and you feel that this manuscript is now acceptable for publication, you may indicate that here to bypass the “Comments to the Author” section, enter your conflict of interest statement in the “Confidential to Editor” section, and submit your "Accept" recommendation.

Reviewer #1: All comments have been addressed

Reviewer #2: All comments have been addressed

2. Is the manuscript technically sound, and do the data support the conclusions?

Reviewer #1: Yes

Reviewer #2: Yes

3. Has the statistical analysis been performed appropriately and rigorously? 

Reviewer #1: Yes

Reviewer #2: Yes

4. Have the authors made all data underlying the findings in their manuscript fully available?

Reviewer #1: Yes

Reviewer #2: Yes

5. Is the manuscript presented in an intelligible fashion and written in standard English?

Reviewer #1: Yes

Reviewer #2: Yes

6. Review Comments to the Author

Reviewer #1: (No Response)

Reviewer #2: I appreciate the authors' answers, which helped me clarify some parts of the manuscript. The authors did a great job implementing the comments, and the manuscript gained clarity and structure.

I suggest adding some references in the statistical part of the method section.

7. PLOS authors have the option to publish the peer review history of their article (what does this mean?). If published, this will include your full peer review and any attached files.

Reviewer #1: No

Reviewer #2: No

---

## [Author Response · Author response to Decision Letter 1]

24 Jul 2024

Dear Academic Editor and Reviewers,

We would like to thank the Academic Editor and the two Reviewers for the additional comments, that improved again this manuscript. We are grateful for their contributions.

Best wishes,

T. Trombetta, B. Mostajir, J. Courboulès, M. Protopapa, S. Mas, N. Aberle and F. Vidussi

---

## [Editor Report · Decision Letter 2]

25 Jul 2024

Warming and trophic structure tightly control phytoplankton bloom amplitude, composition and succession.

PONE-D-24-12240R2

Dear Dr. Trombetta,

We’re pleased to inform you that your manuscript has been judged scientifically suitable for publication and will be formally accepted for publication once it meets all outstanding technical requirements.

Kind regards,

Hans G. Dam, Ph. D.

Academic Editor

PLOS ONE
---

## [Editor Report · Acceptance letter]

6 Aug 2024

PONE-D-24-12240R2 

PLOS ONE

Dear Dr. Trombetta, 

I'm pleased to inform you that your manuscript has been deemed suitable for publication in PLOS ONE. Congratulations! Your manuscript is now being handed over to our production team.

Kind regards, 

on behalf of

Dr. Hans G. Dam 

Academic Editor

PLOS ONE